# MEMORY OF UNIMAGINABLE OUTCOMES IN EXPERIENCE REPLAY

## ABSTRACT

Model-based reinforcement learning (MBRL) applies a single-shot dynamics model to imagined actions to select those with best expected outcome. The dynamics model is an unfaithful representation of the environment physics, and its capacity to predict the outcome of a future action varies as it is trained iteratively. An experience replay buffer collects the outcomes of all actions executed in the environment and is used to iteratively train the dynamics model. With growing experience, it is expected that the model becomes more accurate at predicting the outcome and expected reward of imagined actions. However, training times and memory requirements drastically increase with the growing collection of experiences. Indeed, it would be preferable to retain only those experiences that could not be anticipated by the model while interacting with the environment. We argue that doing so results in a lean replay buffer with diverse experiences that correspond directly to the model's predictive weaknesses at a given point in time.

We propose strategies for: i) determining reliable predictions of the dynamics model with respect to the imagined actions, ii) retaining only the unimaginable experiences in the replay buffer, and iii) training further only when sufficient novel experience has been acquired. We show that these contributions lead to lower training times, drastic reduction of the replay buffer size, fewer updates to the dynamics model and reduction of catastrophic forgetting. All of which enable the effective implementation of continual-learning agents using MBRL.

## 1 INTRODUCTION

Model-Based Reinforcement Learning (MBRL) is attractive because it tends to have a lower sample complexity compared to model-free algorithms like Soft Actor Critic (SAC) (Haarnoja et al. (2018)). MBRL agents function by building a model of the environment in order to predict trajectories of future states based off of imagined actions. An MBRL agent maintains an extensive history of its observations, its actions in response to observations, the resulting reward, and new observation in an experience replay buffer. The information stored in the replay buffer is used to train a single-shot dynamics model that iteratively predicts the outcomes of imagined actions into a trajectory of future states. At each time step, the agent executes only the first action in the trajectory, and then the model re-imagines a new trajectory given the result of this action (Nagabandi et al. (2018)). Yet, many real-world tasks consist in sequences of subtasks of arbitrary length accruing repetitive experiences, for example driving over a long straight and then taking a corner. Capturing the complete dynamics here requires longer sessions of continual learning. (Xie & Finn (2021))

Optimization of the experience replay methodology is an open problem. Choice of size and maintenance strategy for the replay buffer both have considerable impact on asymptotic performance and training stability (Zhang & Sutton (2017)). From a resource perspective, the size and maintenance strategy of the replay buffer pose major concerns for longer learning sessions.

The issue of overfitting is also a concern when accumulating similar or repetitive states. The buffer can become inundated with redundant information while consequently under-representing other important states. Indefinite training on redundant data can result in an inability to generalize to, or remember, less common states. Conversely, too small a buffer will be unlikely to retain sufficient relevant experience into the future. Ideally, a buffer's size would be the exact size needed to capture sufficient detail for all relevant states (Zhang & Sutton (2017)). Note that knowing a priori all relevant states is unfeasible without extensive exploration.

We argue that these problems can be subverted by employing a strategy that avoids retaining experiences that the model already has sufficiently mastered. Humans seem to perform known actions almost unconsciously (e.g., walking) but they reflect on actions that lead to unanticipated events (e.g. walking over seemingly solid ice and falling through). Such is our inspiration to attempt to curate the replay buffer based on whether the experiences are predictable for the model.

Through this work, we propose techniques to capture both common and sporadic experiences with sufficient detail for prediction in longer learning sessions. The approach comprises strategies for: i) determining reliable predictions of the dynamics model with respect to the imagined actions, ii) retaining only the unimaginable experiences in the replay buffer, iii) training further only when sufficient novel experience has been acquired, and iv) reducing the effects of catastrophic forgetting. These strategies enable a model to self-manage both its buffer size and its decisions to train, drastically reducing the wall-time needed to converge. These are critical improvements toward the implementation of effective and stable continual-learning agents.

Our contributions can be summarized as follows: i) contributions towards the applicability of MBRL in continual learning settings, ii) a method to keep the replay buffer size to a minimum without sacrificing performance, iii) a method that reduces the training time. These contributions result in keeping only useful information in a balanced replay buffer even during longer learning sessions.

## 2 RELATED WORK

Compared to MFRL, MBRL tends to be more sample-efficient (Deisenroth et al. (2013)) at a cost of reduced performance. Recent work by Nagabandi et al. (2018) showed that neural networks efficiently reduce sample complexity for problems with high-dimensional non-linear dynamics. MBRL approaches need to induce potential actions which will be evaluated with a dynamics model to choose those with best reward. Random shooting methods artificially generate large number of actions (Rao (2010)) and model predictive control (MPC) can be used to select actions (Camacho et al. (2004)). Neural networks (NNs) are a suitable alternative to families of equations used to model the environment dynamics in MBRL (Williams et al. (2017)). But, overconfident incorrect predictions, which are common in DNNs, can be harmful. Thus, quantifying predictive uncertainty, a weakness in standard NN, becomes crucial. Ensembles of probabilistic NNs proved a good alternative to Bayesian NNs in determining predictive uncertainty (Lakshminarayanan et al. (2016)). Furthermore, an extensive analysis about the types of model that better estimate uncertainty in the MBRL setting favored ensembles of probabilistic NNs (Chua et al. (2018)). The authors identified two types of uncertainty: aleatoric (inherent to the process) and epistemic (resulting from datasets with too few data points). Combining uncertainty aware probabilistic ensembles in the trajectory sampling of the MPC with a cross entropy controller the authors demonstrated asymptotic performance comparable to SAC but with sample efficient convergence. The MPC, however, is still computationally expensive (Chua et al. (2018); Zhu et al. (2020)). Quantification of predictive uncertainty serves as a notion of confidence in an imagined trajectory. Remonda et al. (2021), used this concept to prevent unnecessary recalculation, effectively using sequences of actions the model is confident in and reducing computations. Our approach also seeks to determine reliable predictions of the dynamics model with respect to the imagined actions, but as a basis to manage growth of the experience replay.
**Use of Experience Replay in MBRL:** While an uncertainty aware dynamics model helps to mitigate the risks of prediction overconfidence, other challenges remain. Another considerable issue when training an MBRL agent is the shifting of the state distribution as the model trains. Experience replay was introduced by Lin (1992), and has been further improved upon. Typically in RL, transitions are sampled uniformly from the replay buffer at each step. Prioritized experience replay (PER) (Schaul et al. (2016)) attempts to make learning more efficient by sampling more frequently transitions that are more relevant for learning. PER improves how the model samples experiences from the already-filled replay buffer, but it does not address how the replay buffer is filled in the first place. In addition, neither work addresses the importance of the size of the replay buffer as a hyperparameter (Zhang & Sutton (2017)). Our method attempts to balance the replay buffer by only adding experiences that should improve the future prediction capacity and keeps the training time bounded to a minimum.
**Task Agnostic Continual Learning:** The context of our work originates in tasks consisting in combinations of possibly repetitive subtasks of arbitrary length. In the terminology of Normandin et al. (2021), we aim for continuous task-agnostic continual reinforcement learning. Meaning that the

task boundaries are not observed and transitions may occur gradually (Zeno et al. (2021)). In our case, the task latent variable is not observed and the model has no explicit information about task transitions. In such context, a continual learner can be seen as an autonomous agent learning over an endless stream of tasks, where the agent has to: i) continually adapt in a non-stationary environment, ii) retain memories which are useful, iii) manage compute and memory resources over a long period of time ( Khetarpal et al. (2020), Thrun (1994)). Our proposed strategies satisfy these requirements. Matiisen et al. (2020) address the issue of retaining useful memories in a curriculum learning setting by training a "teacher" function that mandates a learning and re-learning schedule for the agent assuming that the agent will not frequently revisit old experiences/states and will eventually forget them. Ammar et al. (2015) focus on agents that acquire knowledge incrementally by learning multiple tasks consecutively over their lifetime. Their approach rapidly learns high performance safe control policies based on previously learned knowledge and safety constraints on each task, accumulating knowledge over multiple consecutive tasks to optimize overall performance. Bou Ammar & Taylor (2014) developed a lifelong learner for policy gradient RL. Instead of learning a control policy for a task from scratch, they leverage on the agent's previously learned knowledge. Knowledge is shared via a latent basis that captures reusable components of the learned policies. The latent basis is then updated with newly acquired knowledge. This resulted in an accelerated learning of new task and an improvement in the performance of existing models without retraining on their respective tasks. With our method, we imbue the RL agent with the ability to self-evaluate and decide in real-time if it has sufficiently learned the current state. Unlike the method presented by Matiisen et al. (2020), our method requires no additional networks to be trained in parallel.

Xie & Finn (2021) identified two core challenges in the lifelong learning setting: enabling forward transfer, i.e. reusing knowledge from previous tasks to improve learning new tasks, and to improve backward transfer which can be seen as avoiding catastrophic forgetting (Kirkpatrick et al. (2017)). They developed a method that exploits data collected from previous tasks to cumulatively grow the agent's skill-set using importance sampling. Their method requires the agent to know when the task changes whereas our method does not have this constrain. Additionally, they focus in forward transfer only. Our method addresses both forward and backward transfer.

# 3 PRELIMINARIES

At each time $t$, the agent is at a state $s_t \in S$, executes an action $a_t \in A$ and receives from the environment a reward $r_t = r(s_t, a_t)$ and a state $s_{t+1}$ according to some environment transition function $f : S \times A \to S$. RL consists in training a policy towards maximizing the accumulated reward obtained from the environment. The goal is to maximize the sum of discounted rewards $\sum_{i=t}^{\infty} \gamma^{(i-t)} r(s_i, a_i)$, where $\gamma \in [0, 1]$. Instead, given a current state $s_t$, MBRL artificially generates a huge amount of potential future actions, for instance using random shooting ( Rao (2010)) or cross entropy( Chua et al. (2018)). Clarification of these methods is beyond the scope of this paper; we defer the interested reader to the bibliography. MBRL attempts to learn a discrete time dynamics model $\hat{f} = (s_t, a_t)$ to predict the future state $\hat{s}_{t+\Delta_t}$ of executing action $a_t$ at state $s_t$. To reach a state into the future, the dynamics model *iteratively* evaluates sequences of actions, $a_{t:t+H} = (a_t, \ldots, a_{t+H-1})$ over a longer horizon $H$, to maximize their discounted reward $\sum_{i=t}^{t+H-1} \gamma^{(i-t)} r(s_i, a_i)$. These sequences of actions with predicted outcomes are called imagined trajectories. The dynamics model $\hat{f}$ is an inaccurate representation of the transition function $f$ and the future is only partially observable. So, the controller executes only a single action $a_t$ in the trajectory before solving the optimization again with the updated state $s_{t+1}$. The process is formalized in Algorithm 1. The dynamics model $\hat{f}_\theta$ is learned with data $\mathcal{D}_{env}$, collected on the fly. With $\hat{f}_\theta$, the simulator starts and the controller is called to plan the best trajectory resulting in $a^*_{t:t+H}$. Only the first action of the trajectory $a^*_t$ is executed in the environment and the rest is discarded. This is repeated for $TaskHorizon$ number of steps. The data collected from the environment is added to $\mathcal{D}_{env}$ and $\hat{f}_\theta$ is trained further. The process repeats for $NIterations$. Note that generating imagined trajectories requires subsequent calls to the dynamics model to chain predicted future states $s_{t+n}$ with future actions up to the task horizon, and so it is only partially parallelizable.

**Dynamics model.** We use a probabilistic model to model a probability distribution of next state given current state and an action. To be specific, we use a regression model realized using a neural network similar to Lakshminarayanan et al. (2016) and Chua et al. (2018). The last layer of the

model outputs parameters of a Gaussian distribution that models the aleatoric uncertainty (the uncertainty due to the randomness of the environment). Its parameters are learned together with the parameters of the neural network. To model the epistemic uncertainty (the uncertainty of the dynamics model due to generalization errors), we use ensembles with bagging where the members of the ensemble are identical and only differ in the initial weight values. Each element of the ensemble has as input the current state $s_t$ and action $a_t$ and is trained to predict the difference between $s_t$ and $s_{t+1}$, instead of directly predicting the next step. Thus the learning objective for the dynamics model becomes, $\Delta s = s_{t+1} - s_t$. $\hat{f}_\theta$ outputs the probability distribution of the future state $p_{s(t+1)}$ from which we can sample the future step and its confidence $\hat{s}, \hat{s}_\sigma = \hat{f}_\theta(s, [\mathbf{a}])$. Where the confidence $s_\sigma$ captures both, epistemic and aleatoric uncertainty.

---

**Algorithm 1** MBRL

---

Init $\mathcal{D}$ with one iteration of a random controller
**for** Iteration $i = 1$ **to** $NIterations$ **do**
    Train $\hat{f}$ given $\mathcal{D}$
    **for** Time $t = 0$ **to** $TaskHorizon$ **do**
        Get $a^*_{t:t+H}$ from $ComputeOptimalTrajectory(s_t, \hat{f})$
        Execute $a^*_t$ from optimal actions $a^*_{t:t+H}$
        Record outcome: $\mathcal{D} \leftarrow \mathcal{D} \cup \{s_t, a^*_t, s_{t+1}\}$

---

**Trajectory Generation.** Each ensemble element outputs the parameters of a normal distribution. To generate trajectories, P particles are created from the current state, $s_t^p = s_t$, which are then propagated by: $s_{t+1}^p \sim \hat{f}_b(s_t^p, a_t)$, using a particular bootstrap element $b \in \{1, ..., B\}$. Chua et al experimented with diverse methods to propagate particles through the ensemble. The $TS_\infty$ method delivered the best results. It refers to particles never changing the initial bootstrap element. Doing so, results in having both uncertainties separated at the end of the trajectory. Specifically, aleatoric state variance is the average variance of particles of same bootstrap, whilst epistemic state variance is the variance of the average of particles of same bootstrap indexes. We use also $TS_\infty$.

**Control.** To select the best course of action leading to $s_H$, MBRL generates a large number of trajectories $K$ and evaluates them in terms of reward. To find the actions that maximize reward, we used the cross entropy method (CEM) Botev et al. (2013), an algorithm for solving optimization problems based on cross-entropy minimization. CEM gradually changes the sampling distribution of the random search so that the rare-event is more likely to occur and estimates a sequence of sampling distributions that converges to a distribution with probability mass concentrated in a region of near-optimal solutions. Appendix A details the use of CEM to get the optimal sequence of actions $a^*_{t:t+H}$

## 4 TOWARDS CONTINUAL LEARNING

Applying MBRL to a continual learning setting is a promising venue for research. The dynamics model could be constantly improving and adapting dynamically to changes in the environment. Many real-world tasks can be broken in sequences of subtasks of arbitrary length. Capturing the complete dynamics then requires exposure to longer sessions of continual learning. Arbitrarily long repetitive tasks lead to increasing redundancy in the experience replay constantly increasing of the amount of experience collected. These issues hinder the use of MRBL in continual learning settings. **What to add to the replay buffer:** We posit that it would be preferable to retain only those experiences that could not be adequately anticipated by the model during each episode in the environment. Essentially, we would only like to add to the replay buffer observations for which the model issued a poor prediction. On the contrary, we would like to avoid filling the replay buffer or updating the model on observations that the model is good at predicting. We contend that these two elements will lead eventually to a balanced replay buffer, which will contain only relevant observations. This will contribute to the objective of continual learning.

## 5 UARF: UNCERTAINTY AWARE REPLAY FILTERING

Continual learning requires the MBRL agent to adapt in a non-stationary environment, retaining memories that are useful whilst avoiding catastrophic forgetting, and it can manage compute and memory resources over a long period of time ( Khetarpal et al. (2020)). The proposed method, UARF, addresses these issues with a variety of strategies. Algorithm 2 is the main algorithm used to select which observations to append in the replay buffer. The optimal actions $a^*_{t:t+H}$ are computed

by the $ComputeOptimalTrajectory$ function (See Appendix A) given the current state of the environment $s_t$ and $\hat{f}$. The future trajectory and its uncertainty, $p^*_{r(t+1:t+1+H)}$, is then obtained by using $a^*_{t:t+H}$ and $s_t$ with $\hat{f}$. The variable $unreliableModel$ is set to true when the algorithm believes the imagined trajectory not to be trustworthy. Depending on its value, calculation of new trajectories and additions to the replay buffer will be avoided and therefore computation time and size of the replay buffer will be reduced. If $unreliableModel$ is False, the next predicted action is executed in the environment. Subsequent actions from $a^*_{t:t+H}$ are executed until the $unreliableModel$ flag is set to False or the environment reaches $TaskHorizon$ number of steps. The process is repeated for the maximum iterations allowed for the task. After the first action, every time an action $a^*_{t+1:t+H}$ is executed trajectory computation is avoided and this new observation is *not* added to the replay buffer on the basis that the model can already predict its outcome. If $unreliableModel$ is True, the algorithm calculates a new trajectory and adds the current observation to the replay buffer. Hereby, the buffer stores only observations for which the model could not predict (*imagine*) the outcome.

**Trustworthy imagination (Algorithm 2 L:18-21).** The algorithm that assigns a value to $unreliableModel$ is named BICHO. BICHO will essentially return True as long as the reward projected in the future does not differ significantly with respect to the imagined future reward $p^*_r$ and the confidence of the model remains high. BICHO is built assuming that if parts of the trajectory do not vary, their projected reward will be as imagined by the model with some confidence. After calculating a trajectory, the distribution of rewards $p^*_r$ is calculated for H steps in the future. Whereas, at each step of the environment, independent if the recalculation was skipped or not, a new trajectory $p'_r$ of H steps is projected, starting from state $s_t$ which is given by the environment and using actions $a^*_{t+i}$ in the imagined trajectory. We use the Wasserstein distance (Bellemare et al. (2017)) to find how much these two distributions change after each time step in the environment. If the change is $> \beta$ (which is a hyper parameter to tune) then $unreliableModel$ is True. We can control how many steps ahead we would like to compare the two distributions. The comparison is done for just $c$ steps ($< H$), which is a hyper parameter to tune. If they differ significantly, then the trajectory is unreliable. That is, if the projected reward differs from the imagined one the outcome of the actions is uncertain and the trajectory should be recalculated.

Even for a model that has converged, predicting trajectories of great length is impossible. Recalculations inevitably occur at the end of trajectories. Such recalculations do not necessarily represent the appearance of unseen information, but rather a limitation of the successful model in a complex environment. Hence, we would not want to add them to the buffer. The *maximum prediction distance* (MPD) defines a cutoff for a trajectory, and adjusts the strictness of the filtering mechanism. Refer to Appendix E for an extensive analysis.

**Updates on novel information (Algorithm 2 L:24-25)** over-training the dynamics model leads to instabilities due to overfitting. This problem is exacerbated when the replay buffer contains just the minimum essential data. If we only filter the replay buffer, continuously updating the parameters of the dynamics model will eventually lead to overfitting. Instead, our method updates the parameters of the dynamics model only when there is sufficient new information in the replay buffer. We train the dynamics model only when new data exceeds the $new\_data\_threshold$ hyper parameter. For our experiments we set this variable to $0.01$ training only when $1\%$ of the experiences in the replay buffer are new since the last update of the parameters of the dynamics model.

# 6 EXPERIMENTS

The primary purpose of the proposed algorithm is for the resulting replay buffer to retain only relevant, non-redundant, experiences that will be useful for learning the task. We envision applying this method to tasks that require longer training sessions and in continual learning settings. We designed three experimental procedures. The first experiment seeks to establish that our method indeed retains a reduced buffer sufficient for achieving expected rewards when learning a single task throughout long training sessions. To this end, we evaluate the proposed method in benchmark environments for higher number of episodes than in Chua et al. (2018). The second experiment seeks to prove that UARF retains a small number of complementary experiences compared to non-filtering baseline algorithms when training on a sequence of different but related tasks in a continual learning setting. We evaluate our method in a combined task including unseen subtasks. The third experiment seeks to show how UARF addresses the effects of catastrophic forgetting.

---

**Algorithm 2** UARF

---

1: Initialize dynamics model $\hat{f}$ parameters; Initialize replay buffer $\mathcal{D}$ with an iteration of a random controller
2: $unreliableModel = True$ and $trainModel = False$
3: **for** Iteration $l = 1$ **to** $NIterations$ **do**
4:     **if** $trainModel$ **then** Train $\hat{f}$ given $\mathcal{D}$
5:     **for** Time $t = 0$ **to** $TaskHorizon$ **do**
6:         **if** $unreliableModel$ **then**
7:             Get $a^*_{t:t+H}$ from $ComputeOptimalTrajectory\,(s_t, \hat{f})$
8:             Get $p^*_{r(t+1:t+H)}$ given $(s_t, \hat{f}, a^*_{t:t+H})$ // Use $\hat{f}$ to predict H rewards ahead
9:             $i = 0$
10:         **else** $i\ +\!= 1$
11:         Get first action $a_t$ from available optimal actions $a^*_{t:t+H}$
12:         Execute in the environment $a_t$ to obtain $s_{t+1}$ and $r_{t+1}$
13:         Discard first action and keep the rest $a^*_t = a^*_{t+1:t+H}$
14:         Get $p'_{r(t+i+1:t+H)}$ given $(s_t, \hat{f}, a^*_{t+i:t+H})$
15:         // Trustworthy imagination
16:         L = min(H, c - i) // Calculate the number steps ahead to consider
17:         $r\_error = WassersteinDistance(p'_{r(t+i+1:t+i+L)}||p^*_{r(t+1:t+L)})$
18:         **if** $r\_error > \beta$ **then** unreliableModel = TRUE **else** unreliableModel = False
19:         **if** $unreliableModel$ **then**
20:             Record outcome: $\mathcal{D} \leftarrow \mathcal{D} \cup \{s_t, a_t, s_{t+1}\}$
21:             // Updates on novel information
22:             **if** $new\_data\_in\ \mathcal{D} > new\_data\_threshold * length(\mathcal{D})$ **then**
23:                 $trainModel$ = True

---

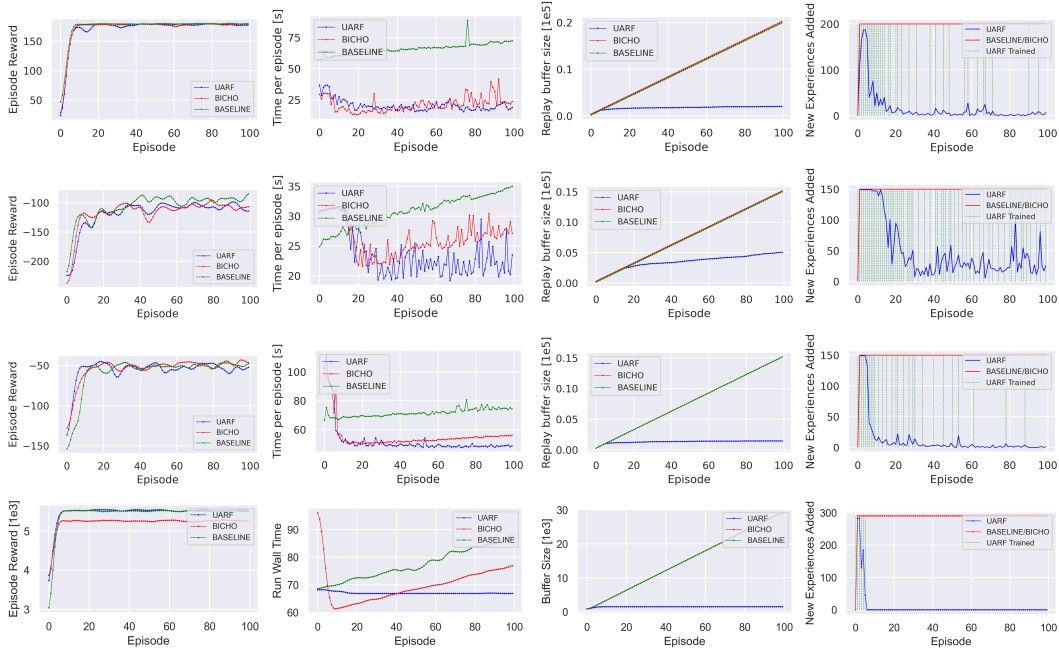

Figure 1: Performance of algorithms (BL: green, BICHO: red, UARF: blue) in (top to bottom) Cartpole, Pusher, Reacher, and Masspoint sector1. From left to right column: episode reward, time per episode (s), cumulative number of observations stored in the replay buffer, new experiences added to the buffer per episode. The rightmost plots illustrate with dashed vertical lines episodes that resulted in UARF updating its model parameters.

## 6.1 E1– CONTINUING TO LEARN A TASK AFTER CONVERGENCE

This experiment is intended to show that our method retains sufficient experience to solve the task while curtailing buffer growth and unnecessary model updates. We intend to prove that this results in

a dramatic reduction in the replay buffer size (which is free of any artificially-imposed limits) while retaining strong performance (per-episode reward) and reducing per-episode wall clock run-time.

We use the MuJoCo (Todorov et al. (2012)) physics engine and environments Cartpole (CP), Pusher (PU) and Reacher (RE) with task length ($TaskH$) and trajectory horizon ($H$) chosen for a valid comparison with Chua et al. (2018). With similar training scenarios, Remonda et al. (2021) trained CP for 30 episodes, PU and RE for 150. Instead, we trained each for 100 episodes. We also included a modified version of the Masspoint environment (Thananjeyan et al. (2020)) (also used in E2). Masspoint is a navigation task in which a point mass navigates to a given goal. It is a 5-dimensional $(x, y, v_x, v_y, \rho)$ state domain. Where $(x, y)$ is the position of the agent, $(v_x, v_y)$ its speed, and $\rho$ is the distance between the agent and the clos-

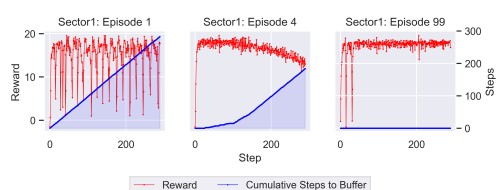

Figure 2: Per-step reward and cumulative steps added to the replay buffer for episodes 1 (left), 4 (middle), and 99 (right) in the Masspoint Sector 1 maneuver. These plots show that UARF adds fewer redundant experiences to the replay buffer as the model converges.

est point to a given path. The agent can exert force in cardinal directions and experiences drag coefficient $\psi$. We use $\psi = 0.6$ and included noise in the starting position. We modified the goal of the agent so that it must move as fast as possible without deviating from a given path. Each task and its complexity is then determined by the geometry of the path to be followed. The reward is calculated as $r = V(1 - |\rho|)$. Where $V$ is the speed of the agent and $\rho$ the distance to the task's path. This experiment used sector1 (Figure in Appendix B) and Hyperparameters shown in Appendix F.

We assess performance in terms of per-episode reward, per-episode wall time, and replay buffer size. We evaluate three algorithms: baseline (BL) is a conventional MBRL (PETS Chua et al. (2018)), BICHO uses functionality to avoid unnecessary computation, and UARF. BICHO and UARF used the same values of $\beta$ and look-ahead, estimated empirically to produce a reasonable balance in terms of per-episode reward and percentage of recalculation. All experiments use random seeds and randomized initial conditions for each run, and ran in workstations with Nvidia 3080TI GPUs.

**Results:** Fig 1 top shows the results obtained in CP. Fig 1-mid-right shows the size of the replay buffer during training. We observe that while the replay buffer keeps grows in the case of BL and BICHO, the size of the buffer derived from UARF is comparably flat: the buffer resulting from UARF is 10x smaller. The training time per episode (Fig 1 mid-left) remains nearly constant and lower for UARF. BL takes substantially longer than both BICHO and UARF to complete an episode. The wall time of both the BL and BICHO exhibit linear growth. It takes longer to update the model as the replay buffer grows linearly. Fig 1-left shows comparable reward per episode for all methods. Results in Fig 1 for PU (row 2), RE (row 3) and Masspoint (bottom) are consistent with those of CP. Fig 2 illustrates the management of buffer growth in Masspoint by showing exactly at which steps experiences are added to the replay buffer during untrained (E1) and trained (E99) episodes. These plots reveal that when the model is untrained, many experiences are added to the buffer throughout the episode. After the model is trained (E99), UARF stops adding experiences to the buffer as the model is able to predict them. Hence, new experiences are deemed redundant and not useful to the model. Results support our claim that UARF obtains a drastically smaller replay buffer that is intelligently populated with only relevant information. This is achieved while maintaining strong performance in the environment compared to BL. Note that while the curves are plotted per episode, it is misleading to assume that all methods converge roughly at the same time. The time per episode in the case of UARF is at least half that of BL for approximately in every environment and it remains stable, while it increases linearly for BL and BICHO with increasing buffer size. The total wall time in average for CP was BL=1.83h, BICHO=0.59h and UARF=0.57h. For PU, it was BL=0.97h, BICHO=0.73h and UARF=0.66h. For RE, it was BL=1.99h BICHO=1.55h and UARF=1.45h, and for MP it was BL=2.15h, BICHO=1.94h and UARF=1.68h.

## 6.2 Ex 2. Continual Learning Experiment

This experiment is set in Task Agnostic Continual Reinforcement Learning (the model is not aware of tasks or task transitions). In this setting, we sought to prove that UARF maintains a leaner and

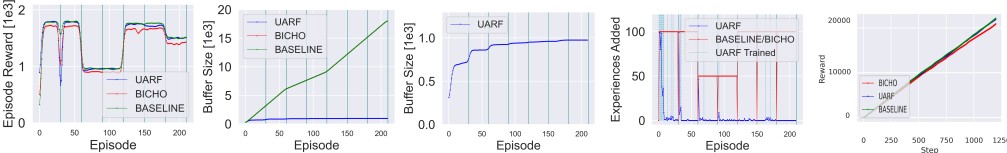

Figure 3: Training performance of Baseline, BICHO and UARF in a Task Agnostic Continual Learning. Models trained on seven maneuvers: corner1, corner1 inverse, chicane, chicane inverse, corner14, corner14 inverse, and straight. Vertical lines indicate a task switch. In the middle-right plot, cyan vertical lines also indicate when UARF triggers a model update. Full-track evaluation in the far-right plot; cumulative reward achieved at each step on the full-track without further training.

more relevant collection of experiences in the replay buffer than do baseline algorithms. These characteristics of the proposed algorithm, we posit, result in strong test performance with less data and greater stability. The existence of these characteristics can be verified by observing (after training) the size of the buffer, the number of experiences from each maneuver present in the buffer, and the performance of the models on the test task. We used the Masspoint racing environment, defining different simple tasks that can be composed to solve a complex, unseen one.

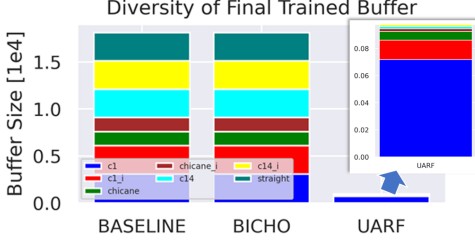

Figure 4: Distribution of experiences from each sub-task in the replay buffers of each algorithm immediately following training. Detail shows a zoomed-in version for UARF.

Each algorithm trains a model on a sequence of seven separate sub-tasks: two corners and their inverses, a chicane and its inverse, and a straight (details illustrated in Appendix B). The models retain their parameters and replay buffers between training on each task individually. After training on the last task, the methods are each tested on the full track, which contains some of the sub-tasks seen during training (colored in the full-track image, Appendix B) and tasks unseen during training (shown in black in the full-track image). The model must remember what it learned by training on each sub-task and apply this knowledge to navigate a more complex, unseen task. All of the algorithms had a virtually unlimited replay buffer size. Each model was trained for 30 episodes on each sub-task and then tested on the test task.

**Results** Figure 3 shows episode reward, wall-time, buffer size during training, and new experiences added to the buffer per episode. Vertical lines illustrate task divisions. High episode reward indicates that each model adequately learns each subtask. UARF maintains almost a constant wall-time, while BL and BICHO increase as experience accumulates. Buffer growth for BL and BICHO is linear, but UARF evidences asymptotic growth (13x smaller) adding no new experiences at the end of training. Figure 3-4 shows the buffer growth of UARF. A larger amount of additions to the replay buffer occur while training the first tasks. Growth slows to a near halt during the last tasks. This is the case for example with the fourth task (chicane inverted). The previous task (chicane) is similar, and the information to solve the previous task is enough that the algorithm does not require a significant amount of new experience to solve chicane inverted. Figure 4 shows the distribution of experiences from each sub-task present in each algorithm's replay buffer immediately following training. BL and BICHO employ a naive approach, resulting in replay buffers with distributions of experience determined exclusively by the length of the various maneuvers. The filtering mechanism of UARF results in a distribution of experience with some maneuvers having limited representation (e.g., the inverse maneuvers) This is because the UARF algorithm intelligently decides to omit redundant experiences from the buffer and leaves only the relevant ones. Figure 3 right shows that all three algorithms result in a model that adequately solves the test task. UARF continues to manage buffer growth while achieving high performance. The results support our initial hypothesis by illustrating clearly the proposed algorithm's propensity to maintain a smaller and more relevant replay buffer while achieving the performance of the baseline in a continual learning setting.

6.3 Ex 3. Catastrophic Forgetting

Our approach helps to mitigate catastrophic forgetting. When using a fixed replay buffer size, it is important to ensure that the appropriate maximum buffer size is chosen (Zhang & Sutton (2017)). If this value is undertuned, important experiences can be jettisoned, and catastrophic forgetting can occur. To illustrate how UARF helps to alleviate this risk, we ran the same experiment shown in section Ex.2 but with a replay buffer of fixed size (5000 samples; roughly 4x the replay buffer size used by UARF in the unlimited size setting). Table 1 compares rewards achieved by each algorithm with both unlimited and fixed buffers. The models were validated on the full track and also on a maneuver that was trained early on in the training process (c1 inverse). Results reveal that with an undertuned fixed buffer size, BL loses about $10\%$ performance both on the full track and on c1 inverse. This is indicative of the fact that the non-filtering algorithms are hitting the buffer size cap, throwing away valuable experiences, and forgetting how to properly solve maneuvers that were trained early on. This impacts performance on the full track as well.

|  | Unlimited Buffer Full Track | Fixed Buffer Full Track | Fixed Buffer c1 inverse First Pass | Fixed Buffer c1 inverse Post-Training |
|---|---|---|---|---|
| BASELINE | 22172 | 20235 | 1787 | 1561 |
| UARF | 21975 | 22102 | 1781 | 1795 |

Table 1: Fixed Buffer Experiment. Results demonstrate susceptibility to catastrophic forgetting when not using UARF. The BL forgets previous maneuvers after the FIFO mechanism of the fixed-size replay buffer eliminates experiences from them with an impact of about $10\%$ in reward.

# 7 Discussion and Conclusion

The results in E1 reveal that continuing to run our algorithm in a repetitive environment with redundant or monotonous actions leads to, in some tasks, no increase in buffer and reduced dynamics model updates. This has the consequence of reduced running and training times, while reducing the effects of catastrophic forgetting and keeping the replay buffer size to a bare minimum. In E2, a continual learning setting, we demonstrated that using our approach leads outcomes with 1/25th of the experiences without performance degradation. UARF effectively deals with an unbounded growth of the replay buffer, which again reduces training time and instabilities. This effect is accentuated when training on a continual learning setting. UARF uses a buffer 43x smaller than the baseline.

The replay buffer is an instrument that makes the use of deep neural networks in RL more stable and it is an essential part in algorithms such as PETs. Such analyses of replay buffer are scarce. But recently, research has turned to analyze the contents and strategies to manage the replay buffer of RL agents Fedus et al. (2020), and also in supervised learning Aljundi et al. (2019). We contribute to such body of work analyzing and offering strategies to manage growth of replay buffer in model based RL. Having managed growth, there are several aspects we would like to turn to in the future: i) identifying task boundary from the novelty of experiences, ii) managing what to forget for limited size buffers, iii) managing what to remember / refresh when a change in task is evident. All this would allow to run agents for arbitrary time without having to deal with size of the buffer and would offer promising opportunities for deploying MBRL in a continual learning setting.

BICHO could be used to prioritize entries in the RB where the model was uncertain. Indeed, prioritized buffer strategies support the usage of experience once it is in the buffer, but as the authors of the PER paper state, strategies for what to add and when (our work) are important open avenues for research. We did not explore our methods in environments where the tasks have interfering dynamics. But, if the dynamics change, poor predictions by the model will result in adding experiences to the replay buffer. What happens if interfering tasks occur permanently is an interesting follow up.

In summary, we proposed strategies that comply with requirements for continual learning. Our approach retains only memories which are useful: it obtains lean and diverse replay buffers capturing both common and sporadic experiences with sufficient detail for prediction in longer learning sessions. Our approach manages compute and memory resources over longer periods: it deals with the unbounded growth of the replay buffer, its training time and instability due to catastrophic forgetting. These results offer promising opportunities for deploying MBRL in a continual learning setting.

## 8 REPRODUCIBILITY STATEMENT

To make our experiments reproducible, we provide the source code in the supplementary material. We include instructions describing how to run all the experiments and to create the images. We include the source code of the proposed algorithms, the MassPoint environment and clear instructions showing how to install extra packages and dependencies needed to reproduce our experiments.

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

## A    OPTIMAL TRAJECTORY GENERATION

Algorithm 3 shows the use of CEM to compute the optimal sequence of actions $a_{t:t+H}^*$.

---
**Algorithm 3** Compute Optimal Trajectory

---
**Input**: $s_{init}$: current state of the environment, dynamics model $\hat{f}$

1: Initialize $P$ particles, $s_\tau^p$, with the initial state, $s_{init}$
2: **for** Actions sampled $a_{t:t+H} \sim CEM(.)$, 1 **to** $CEM Samples$ **do**
3:     Propagate state particles $s_\tau^p$ using TS and $\hat{f}|\{\mathcal{D}, a_{t:t+H}\}$
4:     Evaluate actions as $\sum_{\tau=t}^{t+H} \frac{1}{P} \sum_{p=1}^{P} r(s_\tau^p, a_\tau)$
5:     Update CEM(.) distribution
6: **return** $a_{t:t+H}^*$

---

## B MASS POINT TASKS

Each algorithm trains a model on a sequence of seven separate sub-tasks: two corners and their inverses, a chicane and its inverse, and a straight (Figure 5). The full track contains some of the sub-tasks seen during training (Shown with different colors in the full-track image (Appendix Figure 5) in addition to tasks unseen during training (shown in black in the full-track image).

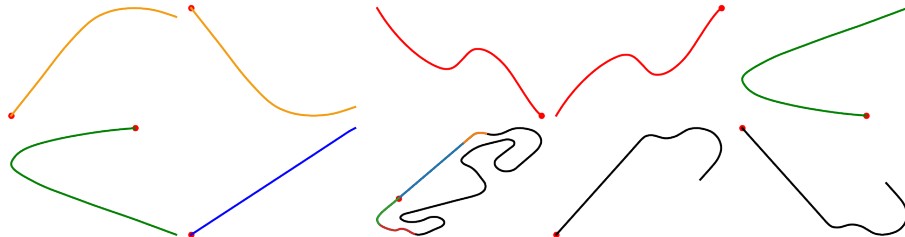

Figure 5: Tasks for the Masspoint environment. The x-axis and the y-axis of each figure represents the x,y coordinates of the path to be followed by the mass point bot. The red dot represents the starting point. Top left-to-right: c1, c1 inverted, chicane, chicane inverted and c14. Bottom left-to-right: c14 inverted, straight, full track (comprising sub-tasks. chicane, c14. straight, c1), sector1 and sector1 inverted

## C ENVIRONMENTS

We evaluate the methods on agents in the MuJoCo Todorov et al. (2012) physics engine. To establish a valid comparison with Chua et al. (2018) we use four environments with corresponding task length ($TaskH$) and trajectory horizon ($H$).

- Cartpole (CP): $S \in \mathbb{R}^4, A \in \mathbb{R}^1, TaskH$ 200, $H$ 25
- Reacher (RE): $S \in \mathbb{R}^{17}, A \in \mathbb{R}^7, TaskH$ 150, $H$ 25
- Pusher (PU): $S \in \mathbb{R}^{20}, A \in \mathbb{R}^7, TaskH$ 150, $H$ 25
- Masspoint: $S \in \mathbb{R}^5, A \in \mathbb{R}^2, TaskH$ 290, $H$ 25

This means that each iteration will run for $TaskH$, task horizon, steps, and that imagined trajectories include $H$ trajectory horizon steps. $S \in \mathbb{R}^i, A \in \mathbb{R}^j$ refers to the dimensions of the environment state consisting in a vector of $i$ components and the action consisting in a vector of $j$ components.

## D EX 2. CONTINUAL LEARNING EXPERIMENT. ADDITIONAL RESULTS

Figure 6 shows additional results with the wall-time during the training process for the continual learning experiment.

## E MAXIMUM PREDICTION DISTANCE

An additional parameter of interest when using UARF is what we call the "maximum prediction distance" or MPD. This parameter operates on the assumption that even for a model that has reached convergence, in some environments, predicting trajectories of great length is impossible. As such, recalculations must inevitably occur at the end of such long trajectories. These recalculations do not necessarily represent the appearance of new, unseen information, but rather a limitation of the successful model in a complex environment. Hence, we would not want to add these experiences to the buffer.

Where we define the cutoff for a trajectory of "great length" can be changed, and it serves to adjust the strictness of UARF's filtering mechanism. For Ex.1 and Ex.2, we chose to set the maximum prediction distance to 1 to ensure the strictest filtering of the replay buffer.

In 7, we evaluate the effect of the MPD on the performance of UARF in the cartpole environment. We were particularly interested in the effect on the rate of recalculation and on the size of the replay buffer. In 7 one can see that the models converge with no issue, but they do differ slightly in the rates of recalculation and buffer filtering. The strictest MPD, MPD=1, results in the leanest buffer, but its recalculation rate is slightly higher than the models with MPD=2 and MPD4.

These results show that the MPD serves as a way to tune the strictness of UARF's buffer filtering mechanism. It would be an area of future research to find the optimal way to tune this parameter automatically throughout training such as to best balance recalculation rate and replay buffer filtering.

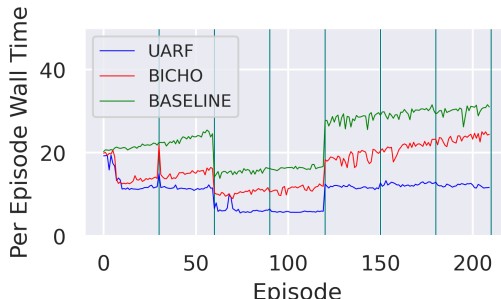

Figure 6: Per episode wall time for the three methods during the training process of Ex.2. Vertical lines indicate task switch points.

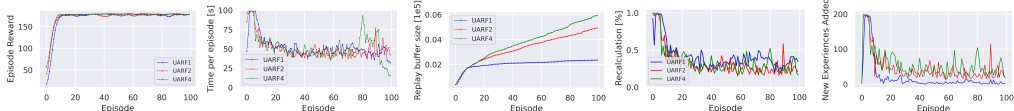

Figure 7: Performance of the examined algorithms in Cartpole using different maximum prediction distances (MPD). The blue line represents UARF with an MPD=1. The red line is UARF with an MPD=2. The green line is UARF with an MPD=4. From left to right column: episode reward, time per episode (s), cumulative number of observations stored in the replay buffer, new experiences added to the buffer per episode.

# F    HYPERPARAMETERS

Table 2 shows the hyper parameters used to train UARF. Look-ahead refers to the number of steps ahead BICHO and UARF are using to asses the quality of the imagined trajectories. $\beta$ controls the sensitivity of BICHO and UARF to inform whether a trajectory is still valid or not. "New Data Train Threshold" refers to the amount of fresh data that must be added to the replay buffer before the UARF algorithm triggers the training of the dynamics model.

|  | Cartpole | Pusher | Reacher | Masspoint |
|---|---|---|---|---|
| Look-Ahead | 10 | 10 | 10 | 10 |
| $\beta$ | 0.005 | 0.005 | 0.005 | 0.5 |
| New Data Threshold | 1% | 1% | 1% | 1% |
| Training episodes | 100 | 100 | 10 | 30/task |
| CEM population | 400 | 500 | 400 | 400 |
| CEM # elites | 40 | 50 | 40 | 40 |
| CEM # iterations | 5 | 5 | 5 | 5 |
| CEM $\alpha$ | 0.1 | 0.1 | 0.1 | 0.1 |
| MPD | 10 | 10 | 10 | 1 |

Table 2: Hyperparameters used for UARF implementation.

