# OpenReview forum: "Memory of Unimaginable Outcomes in Experience Replay"
_ICLR.cc/2023/Conference — Submitted to ICLR 2023_

### Official Review · Reviewer_qNFh · 2022-10-24

**Confidence:** 2
**Correctness:** 3
**Technical Novelty And Significance:** 2
**Empirical Novelty And Significance:** 2
**Recommendation:** 3

**Clarity, Quality, Novelty And Reproducibility:**

* Clarity: I had difficulty understanding the paper. There are quite a few writing issues as discussed above.
* Quality: The proposed methods seem solid. Evaluation seems fair.
* Novelty: To the best of my knowledge, the proposed method for managing replay buffers is novel.
* Reproducibility: Good. Sourcecode is provided.

**Strength And Weaknesses:**

# Strength
* This work addresses an important open problem in how to decide if a transition should be added to the replay buffer. An effective method for maintaining replay buffers can potentially make a big impact as experience replay is a standard building block in many modern deep reinforcement learning agents.
* The experiments show that the proposed method can indeed reduce the memory footprint of the replay buffer and the computational cost while maintaining the agent's performance on the task.

# Weakness
* The paper is challenging to follow. It is hard to capture the main question of interest of this paper at the first pass. The connection to continual learning feels unnatural. Other suggestions on writing improvements are discussed below.
* Section 5 and Algorithm 2 are hard to understand. The authors introduce a lot of notations on-the-fly. I would suggest the authors to define all these notations in clearly all at once. I would also suggest the authors to consider replacing the incomprehensible symbols like $^{*}$ and $'$ with easier-to-interpret ones.
* It will be good to provide a clear description of BICHO. In line 289, it says "BICHO uses functionality to avoid unnecessary computation". This is rather confusing. What functionality? How does it avoid unnecessary computation? Given that BICHO is one of the three methods evaluated in the experiments, I think it is worth spending some space on a clear description of BICHO.
* The proposed method only demonstrates weak evidence in improving the agent's performance on the task in the experiment in Section 6.3. This paper would benefit from a more in-depth evaluation of the proposed method on task performance.

# Questions and minor issues
* On page 1: what does "a single-shot dynamics model" mean?
* The first line of Section 2: The abbreviation "MFRL" shows up without explanation.
* Line 213: "...until the _unreliableModel_ flag is set to False..." I think it should be True but I'm not 100% sure...
* Line 220: What is BICHO precisely? What does "BICHO returns True" mean?
* Line 240: "...(Algorithm 2 L: 24-25)..." There are only 23 lines in Algorithm 2
* Line 300: "Results in Fig 1 for PU, RE, and Masspoint are consistent with those of CP." Not really - BICHO clearly performs worse than BL in Masspoint in the Episode Reward panel.
* Line 386: References to Fedus et al and Aljundi et al should use \citep{}.
* Line 393: What does "RB" stand for?
* Regarding the format: other submissions do not have line number. Could the authors double check if they are using the correct template?

**Summary Of The Paper:**

This work addresses replay buffer management. The authors propose a method that uses model prediction uncertainty to decide the value of each transition. A transition is added to the replay buffer only when its value is above a certain threshold. To combat overfitting and improve computational efficiency, the authors further propose to update the model only when there are enough fresh transitions added to the replay buffer. Experiments in the single-task setting and the task-agnostic continual learning setting demonstrate the efficacy of the proposed methods on reducing the replay buffer size and reducing computational cost. The authors also conduced an experiment to show the potential of the proposed method in mitigating catastrophic forgetting.

**Summary Of The Review:**

I have difficulty understanding this paper thoroughly. I don't think this paper is ready to publish in its current form.

---

### Official Review · Reviewer_3gsD · 2022-10-24

**Confidence:** 4
**Correctness:** 3
**Technical Novelty And Significance:** 2
**Empirical Novelty And Significance:** 2
**Recommendation:** 3

**Clarity, Quality, Novelty And Reproducibility:**

Clarity is generally ok, but certain important details appear to be missing.

I am not aware of other work controlling what is added to a replay buffer based on error so I think the novelty of the general idea is good and worth studying. However, I feel both the approach itself and the evaluation methodology need significant work to be ready for publication at ICLR.

Reproducibility based on the paper would be difficult given missing details, but I can see code is included as supplemental material. I have not looked at the code in any detail.

**Strength And Weaknesses:**

I really like the general idea of this work. Training a model excessively on, and storing, things that are already well-learned is indeed wasteful of computation and memory. Devising approaches to mitigate this is an interesting area for research. I hope the authors continue to work on this topic.

On the other hand, I feel the particular implementation is not very well motivated and the empirical evaluation is not very extensive. For these reasons I don't think the paper is ready for publication at ICLR in its current form.

The notion of model uncertainty used is a little odd to me and doesn't come across as well-motivated, particularly for deciding whether to include transitions in the replay buffer. If the predicted future rewards gradually drift away from their earlier predicted values until the error passes a certain threshold, why does it make sense to assume the particular step at which they passed the threshold is to blame and thus should be used for further model training? Overall, I feel this needs to be better motivated.

Certain details of the algorithm are not clear, for example, "maximum prediction distance" is mentioned as a hyperparameter in line 238, but I can't see it in Algorithm 2 so I had difficulty understanding precisely how it is used. I am also unclear on whether the reward is learned along with the dynamics or provided to the planner explicitly as I couldn't find any mention of reward learning in the paper. If it is provided to the planner, this would have significant implications in particular for the multitask experiments, where I believe essentially all that changes between tasks is the reward. This would also make it disingenuous in my mind to say "the model is not aware of tasks or task transitions" as on line 317 so this point should definitely be clarified.

The empirical results are not very convincing. The experiments in Figure 1 show that by restricting what is added to the replay buffer in the way suggested, replay size remains smaller than if one does not restrict additions, while performance remains similar. However, the obvious baseline of using a fixed-size replay buffer and dropping the oldest item is not included so it's not clearly demonstrated that the particular method used is beneficial. Similarly, it is shown that only replanning when the model is found to be "unreliable" will save computation without sacrificing performance, but this is not compared to simply replanning after a fixed number of steps or any other simple strategy so one cannot say much about the effectiveness of the particular method suggested.

I didn't find the continual learning (task switching) experiments really added much that wasn't already demonstrated in the single-task experiments. Once again we see that the proposed method reduces buffer usage and maintains similar performance, but no simple alternatives for limiting buffer size are evaluated.  One exception is the "catastrophic forgetting" experiment where the effect of limiting buffer size to 5000 is evaluated and shown (with a single random seed?) to improve final performance on past tasks in the continual learning setting. This experiment is a step in the right direction, but I think more needs to be done.

Another issue is that I believe all experiments are run with a single random seed (please correct me if I'm wrong). For the most part, I don't see this affecting the outcome as most claims don't deal with performance. However, it does make it unclear whether the performance benefit compared to a fixed-size buffer in Table 1 is a real effect or just noise. At least this experiment should be done with more random seeds and with confidence bounds.

Another, more minor, issue is the exposition of model-based RL which is currently unclear and misleading. This should be improved to give a more complete picture of the present paper's place in the broader literature, as well as the limitations of the approach. In my mind model-based, RL refers to any reinforcement learning algorithm that learns some model of its environment from experience as an intermediate step to learning, or computing, a good policy. However, this paper makes a number of statements about MBRL that are really about a specific subset of approaches including:
* "An MBRL agent maintains an extensive history of its observations, its actions in response to observations, the resulting reward, and new observation in an experience replay buffer." which refers to algorithms which use experience replay along with a model.
* "At each time step, the agent executes only the first action in the trajectory, and then the model re-imagines a new trajectory given the result of this action." Which seems to refer only to decision time-planning approaches and omits algorithms which use a model in the background for policy learning.
* "the simulator starts and the controller is called to plan the best trajectory resulting in $a^*_{t:t+H}$" seems to refer to specifically open-loop planning.
Note that I don't mean to suggest focusing on a subset of techniques is inherently problematic, only that these things should not be said to characterize MBRL more generally.



Minor Comments and Corrections
==============================
* In the abstract and introduction: what is a "single-shot" dynamics model?
* line 63: MFRL acronym is never actually defined.
* Line 137: $\hat{f}=(s_t,a_t)$ should probably just be $\hat{f}(s_t,a_t)$
* Line 220: Is "BICHO" an acronym? I can't see that it is ever defined.
* Line 220: "BICHO will essentially return True as long as the reward projected in the future does not differ significantly with respect to the imagined future reward...", I think "True" here should be False since BICHO returns true when the model is unreliable.
* Line 107: Backwards quotation marks on "teacher", also elsewhere.
* Line 341: "Figure 3 shows episode reward, wall-time, buffer size...", I don't think wall time is actually included in Figure 3.
* Algorithm 2 (line 17): I think there may be a typo here, as it appears to be comparing rewards with different time indices.


Suggestions
===========
One suggestion for future work in this direction is to work on motivating the problem better, which may in turn lead to a more principled approach. One direction to look at is the use of importance sampling for variance reduction. One can show that lower variance gradient estimates can be achieved by sampling examples with large gradients more often and correcting the bias with importance sampling. This ties in well with the proposed approach of excluding samples which are already well-learned. Perhaps a similar analysis could be done to better justify the proposed approach or derive a more principled way to achieve the same basic goal.

It would be good to have a baseline with restricted buffer size in the experiments in Figure 1 and elsewhere to check whether the particular method of restricted items in the buffer is better than arbitrarily limiting it.

**Summary Of The Paper:**

This paper presents a method to restrict what is stored in a replay buffer used to train a model in order to avoid filling the buffer with things that are already well-learned.  The proposed method is applied with open-loop control using an ensemble of probabilistic models for planning. The method is based on a notion of model uncertainty. Intuitively, if the model predictions are already reliable, in a certain sense, for a given transition, the transition is not added to the buffer. Model uncertainty is simultaneously used to determine when to replan a trajectory during execution, replanning only occurs when model uncertainty becomes high, and the preplanned trajectory is otherwise followed. The particular notion of model uncertainty used appears to be the Wasserstein distance between the predicted distribution of future rewards over some horizon as computed the last time the model predictions were considered "unreliable", and predictions beginning from the current state. Experiments demonstrate that the proposed method reduces buffer size and computation in a number of tasks without compromising performance.

**Summary Of The Review:**

While the general idea is interesting, I feel the particular implementation is not very well motivated and the empirical evaluation is not very extensive or convincing. For these reasons I don't think the paper is ready for publication at ICLR in its current form.

---

### Official Review · Reviewer_SwfR · 2022-10-25

**Confidence:** 3
**Clarity, Quality, Novelty And Reproducibility:** See the above section.
**Correctness:** 2
**Technical Novelty And Significance:** 2
**Empirical Novelty And Significance:** 2
**Recommendation:** 3

**Strength And Weaknesses:**

Strength:

The paper studies an important and interesting topic — I agree that the memory and computation scalability issue is important in an extremely long-time horizon learning setting.

Weaknesses:

----------
Novelty. The paper is mostly combing some intuitive designs together to form an algorithm. I did not see the exactly same algorithm before. But the ideas behind the algorithm are already studied.

----------
Soundness.
The paper is mostly empirical, and the algorithmic design is mostly based on intuition. Though it is fine not to have theoretical support, each design choice should be well-justified. However, many design choices in the paper could be problematically intuitively. Here are some examples.

First, using reward distribution to judge whether a trajectory is novel or not is not well-justified. The reward distribution match may not imply anything about the underlying state-action-state transitions. I cannot say that such a reward-based measure does not work at all, but the paper does not provide any insight into why it works.

Second, it is hard to believe why the method can avoid catastrophic forgetting. Intuitively, if you discard some experiences which have been predicted accurately, the model can still forget those experiences as the sampling distribution shifts (because the policy is changing).

Third, the authors use trajectory optimization and CEM in the MBRL method. There are large amounts of MBRL algorithms. It is better to provide some reasons why the particular method is chosen.

I strongly suggest the authors carefully justify each design choice they used.

----------
Missing a large body of related works.
1. Some simple and theoretically sound memory-saving methods, such as reservoir sampling (I think this should be a simple baseline and it is easy to implement)

2. A body of work saving memory by using theoretically justified approaches in MBRL (e.g. organizing experiences: a deeper look at replay mechanisms for sample-based planning … by Pan et al.)

3. Decision/value-aware robust model-based RL by Amir-massoud et al.

4. There are also papers studying/using techniques to avoid/reduce compounding errors in long-time horizon predictions.

5. It is also good to survey papers about avoiding catastrophic forgetting in a supervised learning setting.

----------
The empirical results should at least justify the key claims of the paper:
1. why the particular uncertainty measure is needed/better than the existing uncertainty measure
2. why the proposed method can avoid catastrophic forgetting

**Summary Of The Paper:**

The paper tries to address the memory scalability issue of MBRL in a very long-time horizon or even infinitely time horizon (say continual learning) settings. An observation is that the model would become more accurate as training proceeds, and some experiences can be discarded. The paper tries to select those experiences which cannot be predicted well by the model to store in the replay buffer. To do this, the paper proposes a method to determine the certainty of the model’s prediction. The method is to measure the Wasserstein distance between the reward distributions of predicted and true trajectories. If the distance is larger some threshold the trajectory would be stored.

To further reduce the computation cost, the authors propose only training the model when sufficient novel experiences are acquired, which is done by simply introducing an additional parameter adjusting how many experiences should be considered as “sufficient”. Experiments are conducted on both standard RL tasks and continual learning tasks.

**Summary Of The Review:**

The paper is most empirical. But it obviously lacks justifications for several important design choices. It also lacks some literature review, and hence some intuitive baselines are not included in the comparison.

---

### Official Review · Reviewer_LPd9 · 2022-10-26

**Confidence:** 3
**Clarity, Quality, Novelty And Reproducibility:** Covered above.
**Correctness:** 2
**Technical Novelty And Significance:** 2
**Empirical Novelty And Significance:** 2
**Recommendation:** 3

**Strength And Weaknesses:**

## Strength

1. The paper studies an important problem - choosing which examples to save to the replay buffer (in the context of MBRL).

## Areas for improvement

1. Writing issues:

    1.1 The paper needs to flow better. It starts with MBRL (which is an important problem to study), then jumps to Replay Buffer (RB) and Continual Learning (CL). The transition from MBRL to RB and CL needs to be clarified. Are RB-related issues the only challenges in MBRL, or why are those issues more important? Similarly, why should one care about CL in the context of MBRL? The authors need to motivate these questions better.

    1.2 Related work needs references to approaches that select what examples are retained/sampled from the replay buffer in the context of CL methods. Given that the paper's primary focus is on reducing the size of the replay buffer and determining what examples/transitions to store, referencing such works is helpful.

    1.3  Typos (like MRBL in line 192)

    1.4 Need to include details like the number of seeds/trials for experiments.

    1.5 Several phrases are not explained, e.g., "complementary experiences" in line 256,

2. Experiments

2.1 Several details are missing or glossed over. E.g., in line 254, "we evaluate the proposed method in benchmarks environments for higher number of episodes than in Chua et al. (2018)" - why more episodes? Or line 270, "With similar training scenarios, Remonda et al. (2021) trained CP for 30 episodes, PU and RE for 150. Instead, we trained each for 100 episodes." - > why change the number of episodes?

2.2 Several hyperparameters are introduced, but the corresponding ablations/analysis are missing.

2.3 Reporting wall-clock time can be misleading as the wall-clock time depends on the load on the system when the time was recorded. A better metric to report is the number of floating-point operations

2.4 It is not obvious to me what the difference is between BICHO and UARF

2.5 In line 298, the paper notes, "It takes longer to update the model as the replay buffer grows linearly." Is this because the system is trained on the entire replay buffer at every update step? Did the authors consider the variant where a fixed number of data points (respective of the size of the replay buffer) is used for an update?

2.6 In line 263, the paper says, "replay buffer size (which is free of any artificially-imposed limits)" - it is unclear if the baseline approaches need the unlimited buffer size. Maybe a small buffer size (equal to the buffer size used by the proposed method) is sufficient for solving the task, and the extra examples are unnecessary. My suspicion grows stronger when I look at the first column of figure 1, where the models often reach quite close to the convergence performance in a few episodes. Still, the training seems to have continued to inflate the size of the replay buffer for the baseline methods. I would also like to see an ablation where the size of the replay buffer (for the baselines) is fixed, and older entries are thrown away in FIFO order as new entries become available.

2.7 Regarding "time per episode," I understand that the baselines would take longer because the replay buffers are larger (read 2.5 for a query on that), but this effect should not kick in when the number of episodes is very small. e.g., in Figure 1, for 0 episodes, I expect all the methods to take the same amount of time (maybe the proposed method takes a bit longer due to the computation of some variables). Still, the PETS baseline is taking much longer, even with 0 episodes. Why is that the case? Isn't the proposed method using the PETS model as the underlying MBRL method?

2.8 In Section 6.2, the tasks considered are basically the same line-following task. Given that the agent has access to the distance from the closest point on the path, there is not much distinction between the tasks. It would not be surprising that an agent that (relatively) easily generalize across different task instances (presented as different tasks).

2.9 The previous question (2.6) about the replay buffer size also applies here. It must be clarified if the baseline needs large buffers to solve the task.

2.10 Generally, the paper needs to consider more complex tasks (with longer episode lengths) if the claims are about improving performance in the lifelong learning setup.

**Summary Of The Paper:**

The paper proposes approaches for determining when to add samples to replay buffer (in context of MBRL). It considers experiments on simple environments to show that the proposed method can perform at par with the baselines while keeping the size of the replay buffer small.

**Summary Of The Review:**

At this point, I think the paper needs several updates to both the writing and the experiments. I encourage the authors to point out mistakes / oversights in my review and I am open to changing my views and scores.

---

### Decision · Program_Chairs · 2023-01-20

**Decision:**

Reject

**Justification For Why Not Higher Score:**

This paper has a number of issues regarding clarity, quality, and novelty and does not meet the bar for publication at ICLR.

**Justification For Why Not Lower Score:**

N/A

**Metareview: Summary, Strengths And Weaknesses:**

This paper proposes a method to limit replay buffer size for training a model for use in MPC, by only adding data to the replay buffer when it is unexpected/difficult to predict. The method, UARF, is evaluated in a few continuous control environments and in some continual learning settings.

While the reviewers felt the problem that the paper studied was important, they also had concerns about the quality and clarity of the writing, lack of important details, insufficient literature review, the rigor of the experiments, and the motivation for the approach. The reviewers have provided many extensive suggestions for how to improve the paper and I would encourage the authors to use this feedback in revising this work going forward.

**Summary Of Ac-Reviewer Meeting:**

N/A